# An Approach for Approximating Analytical Solutions of the Navier-Stokes Time-Fractional Equation Using the Homotopy Perturbation Sumudu Transform's Strategy

**Sajad Iqbal** [1,2,*] and **Francisco Martínez** [3,*]

1 Department of Mathematics, Jiangsu University, Zhenjiang 212013, China
2 Institute of Applied System Analysis, Jiangsu University, Zhenjiang 212013, China
3 Department of Applied Mathematics and Statistics, Technological University of Cartagena, 30203 Cartagena, Spain
* Correspondence: sajad_iqbal@ujs.edu.cn (S.I.); f.martinez@upct.es (F.M.)

**Abstract:** In this study, we utilize the properties of the Sumudu transform (SuT) and combine it with the homotopy perturbation method to address the time fractional Navier-Stokes equation. We introduce a new technique called the homotopy perturbation Sumudu transform Strategy (HPSuTS), which combines the SuT with the homotopy perturbation method using He's polynomials. This approach proves to be powerful and practical for solving various linear and nonlinear fractional partial differential equations (FPDEs) in scientific and engineering fields. We demonstrate the efficiency and simplicity of this method through examples, showcasing its ability to approximate solutions for FPDEs. Additionally, we compare the numerical results obtained using this technique for different values of alpha, showing that as the value moves from a fractional order to an integer order, the solution becomes increasingly similar to the exact solution. Furthermore, we provide the tabular representations of the solution for each example.

**Keywords:** homotopy perturbation strategy; Sumudu transform; approximate solution; fractional Caputo derivatives

## 1. Introduction

Fractional calculus (Fc), an extension of traditional integer-order calculus, has gained significant attention across various scientific and engineering disciplines due to its capability to model complex phenomena with long-range memory and anomalous diffusion. This mathematical discipline dates back to the 17th century, with pioneers like L'Hôpital and Leibniz exploring the idea of Fractional derivative (Fd). However, it was Liouville and Riemann in the late 19th century who laid the rigorous foundations of Fc [1,2]. Fc finds applications in various scientific and engineering fields, including physics, engineering, finance, biology, and signal processing. It allows the modeling of complex phenomena with long-range memory and anomalous diffusion, which cannot be adequately captured by traditional integer-order calculus [1,3–5]. Fc has emerged as a robust mathematical framework that extends the conventional integer-order calculus to accommodate derivatives and integrals of non-integer orders. This extension allows for the description and analysis of complex dynamical systems and processes that exhibit behaviors beyond the scope of traditional calculus. Over the past few decades, Fc has found applications in various scientific and engineering disciplines, such as physics, biology, economics, and engineering, enabling researchers to model intricate phenomena with long-range memory and anomalous diffusion [3,6–8].

The study of FPDEs has gained significant attention in recent years. These equations are frequently encountered in various fields such as fluid mechanics, viscoelasticity, biology,

engineering, and physics [9,10]. However, most of these equations do not have exact analytical solutions, necessitating the use of approximation and numerical techniques. Several numerical methods have been developed to tackle these equations, including the Adomain decomposition method [11], Homotopy analysis method [12], Variational iteration method [13], and Homotopy perturbation method [14–17]. The homotopy perturbation method [18] proposed by He, has proven to be a useful tool for obtaining exact and approximate solutions for both linear and nonlinear FPDEs.

One of the main advantages of the homotopy perturbation method is that it does not require a small parameter or linearization. The solution procedure is straightforward and only a few iterations are needed to achieve highly accurate solutions that are valid for the entire solution domain. Moreover, the solution is expressed as the summation of an infinite series, which is expected to converge to the exact solution. The Sumudu transform (SuT) introduced by Watugala in 1994, has emerged as a valuable mathematical tool for transforming a wide range of integral and DEs into algebraic equations and thus simplifying their solution process [19]. This transform has proven particularly useful in the context of Fc, where it enables the conversion of FDEs into algebraic forms, facilitating the application of various solution methods. The SuT has gained prominence recently due to its versatility and effectiveness in handling complex mathematical problems across different scientific and engineering disciplines. The SuT, known for its scale and unit-preserving properties, eliminates the need for introducing a new frequency domain. It has been demonstrated that the SuT maintains the units of the original problem, making it a valuable tool for solving problems without relying on the frequency domain [20]. One of the notable strengths of the SuT is its applicability to problems involving physical dimensions. This transform, being linear in nature, possesses the ability to preserve linear functions, thereby ensuring that units remain unchanged. This characteristic is particularly advantageous when solving problems that involve physical dimensions [19,21]. When integrated with the HPS, the resulting HPSuTS offers a comprehensive approach for solving FDEs. The SuT aids in simplifying the Fc operations, and the HPS provides an iterative framework for obtaining accurate approximate solutions. This combination of the SuT and the HPS has gained recognition for its ability to handle complex problems involving Fc, enabling researchers to explore a wide range of scientific and engineering applications [19].

In [22], Hamed, Yousif, and Arbab presented a novel approach for solving Schrödinger space-time fractional equations through the integration of the HPS with the SuT. The researchers aimed to obtain both analytic and approximate solutions for these complex equations. The proposed method demonstrated its efficacy in providing accurate solutions, highlighting its potential in handling the challenges posed by Schrödinger space-time fractional equations. In his work, Khader explored the application of the HPSuTS to solve nonlinear heat-like fractional equations. The study aimed to address the challenges posed by these intricate equations by combining the HPS with the SuT technique. By doing so, the researcher sought to derive approximate solutions that capture the nonlinear dynamics of the equations [23]. By exploring the domain of mathematical challenges, recent studies have showcased innovative approaches to overthrow the complexities of nonlinear FPDEs. Many researchers have solved different equations with the help of HPSuTS, such as heat and wave-like equations [24], Black-Scholes European option pricing equations [25] and references therein.

Recently, El-Shahed and Salem generalized the traditional Navier-Stokes equations by introducing a fractional derivative of order $\alpha$ in place of the classical time derivative. They utilized Laplace transform, Fourier sine transform, and finite Hankel transforms to derive approximate solutions for three specific cases [26]. The most important advantage of using fractional differential equations in these and other applications is their non-local property. It is well known that the integer order differential operator is a local operator but the fractional order differential operator is non-local. This means that the next state of a system depends not only upon its current state but also upon all of its historical states. This is more realistic and it is one of the main reason why fractional calculus has become more and more

popular. However, due to the nonlinearity of the Navier-Stokes Time-Fractional Equation (NS-TFEs), there is currently no universally known method for analytically solving this equations. Obtaining an exact solution for this equations is rare and typically requires assumptions about the fluid's state and consideration of a simple flow pattern configuration. Subsequently, various numerical methods have been employed to study this equation, such as the Adomian decomposition method [27], Homotopy analysis method [28], and Modified Laplace decomposition method [29].

In this paper, we will focus on the unsteady flow of a viscous fluid in a tube. The main objective of this research is to expand the applications of the HPS in combination with the SuT to investigate the approximate solution of NS-TFEs in cylindrical frame of reference. Additionally, to demonstrate the accuracy of this suggested technique, we have compared our results with already established results. The results indicates that as the values of $\alpha$ approaches to 1, the solution becomes more similar to exact solution. Moreover, we have provided the tabular representation of the approximate solution for various fractional order.

The structured of this study is as follows. A comprehensive discussion of the essential concepts of Fc is provided in Section 2. The model formulation and applicability are addressed in detail in Section 3. The implementation of the suggested technique and its numerical solution is discussed in Section 4. Finally, concluding remarks are given in Section 5.

## 2. Revealing Fundamental Notions

Fc, a branch of mathematics extending traditional calculus to non-integer orders, has gained prominence for its ability to model complex phenomena with long-range memory and anomalous diffusion. This section explores the concept of fractional Caputo derivatives (FCd), introduces the SuT as a powerful mathematical tool, and discusses their interplay with related properties.

### 2.1. Definition

A mathematical function known as the "Gamma function" generalizes the factorial sense to real and complex numbers. It is denoted by $\Gamma(z)$, where $z$ is a complex number. The gamma function is defined as an integral over the positive real numbers [1,2,30]:

$$\Gamma(z) = \int_0^\infty \eta^{z-1} e^{-\eta} \, d\eta, \tag{1}$$

where the real part of $z$ must be greater than zero for the integral to converge.

### 2.2. Definition

The Riemann–Liouville (R–L) integral operator of fractional order is a mathematical operation that extends the concept of integration to non-integer orders. Given a real function $f(\xi)$ and a parameter $\alpha > 0$, the R–L integral operator of fractional order $_0I^\alpha$ is defined as follows [2,30,31]:

$$_0I^\alpha f(\xi) = \frac{1}{\Gamma(\alpha)} \int_0^\xi (\xi - \eta)^{\alpha-1} f(\eta) d\eta,$$

where $\Gamma(\alpha)$ is the gamma function, and the integral is taken from $\eta = 0$ to $\eta = \xi$.

2.2.1. Properties of Riemann–Liouville Fractional Integral Operator
2.2.2. Linearity: [20,30,31]

For any real constants $a$ and $b$, and functions $f(\xi)$ and $g(\xi)$, we have

$$_0I^\alpha(af(\xi) + bg(\xi)) = a\,_0I^\alpha f(\xi) + b\,_0I^\alpha g(\xi). \tag{2}$$

### 2.2.3. Scaling Property: [20,30,31]

$$_0I^\alpha(\xi^\beta f(\xi)) = \frac{\Gamma(\beta+1)}{\Gamma(\beta+\alpha+1)}\, _0I^{\alpha+\beta}f(\xi), \quad \text{where} \;\; \beta > -1 \tag{3}$$

### 2.2.4. Connection with Derivatives

If $f(\xi)$ has $n$ continuous derivatives on $(0,\xi)$ and $n-1$ continuous derivatives at $\xi = 0$, then $_0I^n f(\xi)$ is the $n$th derivative of the $n$th R–L integral of fractional order of $f(\xi)$ [1,2,30].

### 2.2.5. Inverse Property

The R–L integral of fractional order is invertible, and its inverse is given by [1,2,30],

$$_0I^{-\alpha}f(\xi) = \frac{1}{\Gamma(\alpha)}\int_\xi^\infty (\eta - \xi)^{\alpha-1}f(\eta)d\eta, \quad \text{where} \quad \alpha > 0 \tag{4}$$

### 2.2.6. Composition Rule: [2,30]

If $\alpha > \beta > 0$, then

$$_0I^\alpha(_0I^\beta f(\xi)) = {}_0I^{\alpha+\beta}f(\xi) \tag{5}$$

The R–L integral of fractional order is a fundamental tool in Fc, allowing the extension of traditional calculus concepts to non-integer orders and enabling the study of various complex systems and phenomena.

### 2.3. Definition

The FCd is a mathematical notion that extends the concept of classical derivative to fractional orders. It is often used in Fc to describe the behavior of functions involving Fd. The FCd of a function $f(\xi)$ of order $\alpha$, denoted by ${}^C\mathcal{D}^\alpha f(\xi)$, is defined as follows [2,30]:

$$^C\mathcal{D}^\alpha f(\xi) = \frac{1}{\Gamma(n-\alpha)}\int_a^\xi (\xi-\eta)^{n-\alpha-1}\frac{d^n}{d\eta^n}f(\eta)\,d\eta, \quad n-1 < \alpha \le n \tag{6}$$

where $n$ is the smallest integer greater than $\alpha$, $a$ is a lower limit of integration, and $\Gamma$ represents the gamma function. The FCd has an important relation to ordinary derivatives. When we take $\alpha$ as a positive integer, i.e., $\alpha = n$. In this case, the FCd reduces to the $n$th ordinary derivative:

$$^C\mathcal{D}^n f(\xi) = \frac{d^n}{dx^n}f(\xi).$$

However, for non-integer values of $\alpha$, the FCd involves both the fractional order and an integral of the ordinary derivatives. This integral term accounts for the "memory" of the function $f(\xi)$, capturing the effect of past values of the function up to $n$-th order in the fractional differentiation process.

### 2.4. Definition

The Mittag-Leffler function (M-LF) is a special function that arises in various areas of mathematics, including Fc, complex analysis, and probability theory. It is represented as $\mathcal{E}_{\alpha,\beta}(z)$ and is defined by the following infinite series [2,30]:

$$\mathcal{E}_{\alpha,\beta}(z) = \sum_{m=0}^\infty \frac{z^m}{\Gamma(\alpha m + \beta)}, \tag{7}$$

where $z$ is a complex number, $\alpha$ and $\beta$ are real or complex parameters, and $\Gamma$ is the gamma function. The M-LF generalizes the exponential function and is closely related to Fc. It exhibits behavior that interpolates between exponential decay and algebraic decay, making it suitable for modeling processes with memory or anomalous diffusion.

*2.5. Some Properties of the Mittag-Leffler Function*

2.5.1. Fractional Derivatives: [2,30]

The M-LF is often used to represent Fd in Fc. Specifically, $\mathcal{D}^{\alpha}\mathcal{E}_{\alpha,\beta}(z) = z^{-\alpha}\mathcal{E}_{\alpha,\alpha}(-z)$, where $\mathcal{D}^{\alpha}$ represents a Fd of order $\alpha$.

2.5.2. Series Convergence: [2,30]

The M-LF converges for all complex $z$ and $\alpha, \beta$ such that $\text{Re}(\alpha) > 0$ and $\text{Re}(\beta) \geq 0$.

2.5.3. Special Cases: [2,30]

When $\alpha = 1$ and $\beta = 1$, the M-LF reduces to the ordinary exponential function, $\mathcal{E}_{1,1}(z) = e^{z}$.

*2.6. Definition*

The SuT [19,22,32] of a function $f(\eta)$ defined for $\eta \geq 0$ is denoted as $F_s(v)$. It involves integrating $f(\eta)$ multiplied by an exponential term $\exp\left(-\frac{\eta}{v}\right)$ over the entire non-negative time domain:

$$\mathscr{S}[f(\eta)](v) = F_s(v) = \int_0^{\infty} \frac{1}{v}\exp\left(-\frac{\eta}{v}\right)f(\eta)d\eta, \tag{8}$$

This transform provides insight into the function's behavior in a new parameter $v$-domain, aiding in the solution of various mathematical problems, including integral and differential equations, with applications across diverse fields. Now, let us recall some interesting special properties of the Sumudu Transform that are necessary in the further development of this study.

2.6.1. Constant Function

The SuT of the constant function $f(\eta) = 1$ is equal to 1 [33,34]:

$$\mathscr{S}[1](v) = 1.$$

2.6.2. Exponential Function: [33,34]

For positive $n$, the SuT of the function $f(\eta) = \eta^n$ is given by:

$$\mathscr{S}[\eta^n](v) = v^n\Gamma(n+1).$$

2.6.3. Exponential Decay: [33,34]

The SuT of the exponential decay function $f(\eta) = e^{a\eta}$ is given by:

$$\mathscr{S}[e^{a\eta}](v) = \frac{1}{1 - av}.$$

2.6.4. Linearity: [25,33,34]

The SuT obeys the linearity property:

$$\mathscr{S}[\alpha f(\xi) + \beta g(\xi)](v) = \alpha\mathscr{S}[f(\xi)](v) + \beta\mathscr{S}[g(\xi)](v).$$

For further properties and details about the SuT, see [20,31] and references therein. These properties are useful tools in applying the SuT to various mathematical problems and functions.

2.6.5. Sumudu Transform and Fractional Caputo Derivative

The SuT of the FCd $^{C}\mathcal{D}^{\alpha}f(\eta)$, where $\alpha$ is within the range $(m - 1 < \alpha \leq m)$, is denoted as $\mathscr{S}[^{C}\mathcal{D}^{\alpha}f(\eta)]$ and is calculated as follows [33,34]:

$$\mathcal{S}[^{C}\mathcal{D}^{\alpha}f(\eta)](v) = v^{-\alpha}\mathscr{S}[f(\eta)](v) - \sum_{m=0}^{n-1}v^{-\alpha+m}f^{(m)}(0^+)$$

Here, $\mathscr{S}[f(\eta)]$ is the SuT of the function $f(\eta)$, $f^{(m)}(0^+)$ represents the $m$-th derivative of $f(\eta)$ evaluated at $\eta = 0^+$. This SuT of the FCd aids in analyzing and solving FDEs involving the FCd, providing insights into the transformed function's properties and contributing to the mathematical solutions of various problems.

In the subsequent section, we will provide a comprehensive explanation of how the SuT and the HPS can be utilized to address FDEs in detail.

### 3. Homotopy Perturbation Sumudu Transform Strategy

In this section, we present a comprehensive formulation of a novel approach termed "Homotopy Perturbation Sumudu Transform Strategy" (HPSuTS). This approach is applied to address complex FDEs, which have significant applications in various scientific and engineering domains.

The problem at hand is represented by the following equation:

$$^{\mathcal{C}}\mathcal{D}^{\alpha}u(\xi,\eta) + \Re[u(\xi,\eta)] + \mathcal{N}[u(\xi,\eta)] = \Psi(\xi,\eta), \tag{9}$$

with the initial condition (I.C):

$$u(\xi,0) = \Phi(\xi), \tag{10}$$

where $u(\xi,\eta)$ represents a function of two variables, $^{\mathcal{C}}\mathcal{D}^{\alpha} = \frac{\partial^{\alpha}}{\partial\eta^{\alpha}}$ denotes a FCd of order $\alpha \in (0,1]$, $\Re$ signifies the second part of a linear operator, $\mathcal{N}$ represents a non-linear operator, and $\Psi(\xi,\eta)$ is a non-homogeneous term. Applying the linearity of the SuT, the equation can be rewritten as:

$$\mathscr{S}[^{\mathcal{C}}\mathcal{D}^{\alpha}u(\xi,\eta)](v) + \mathscr{S}[\Re[u(\xi,\eta)]](v) + \mathscr{S}[\mathcal{N}[u(\xi,\eta)]](v) = \mathscr{S}[\Psi(\xi,\eta)](v). \tag{11}$$

Alternatively, the equation can be expressed as:

$$\mathscr{S}[^{\mathcal{C}}\mathcal{D}^{\alpha}u(\xi,\eta)](v) = \mathscr{S}[\Psi(\xi,\eta)](v) - \mathscr{S}[\Re[u(\xi,\eta)]](v) - \mathscr{S}[\mathcal{N}[u(\xi,\eta)]](v) \tag{12}$$

By applying definition (Section 2.6.5), the following expression is derived:

$$\mathscr{S}[u(\xi,\eta)](v) = u(\xi,0) + v^{\alpha}\mathscr{S}[\Psi(\xi,\eta)] - v^{\alpha}\mathscr{S}[\mathcal{N}(u(\xi,\eta)) + \Re(u(\xi,\eta))]. \tag{13}$$

Incorporating the initial condition, the equation becomes:

$$\mathscr{S}[u(\xi,\eta)](v) = \Phi(\xi) + v^{\alpha}\mathscr{S}[\Psi(\xi,\eta)] - v^{\alpha}\mathscr{S}[\mathcal{N}(u(\xi,\eta)) + \Re(u(\xi,\eta))]. \tag{14}$$

Finally, by applying the inverse SuT, we obtained the above expression as:

$$u(\xi,\eta) = \Theta(\xi,\eta) - \mathscr{S}^{-1}\left[v^{\alpha}\mathscr{S}[\mathcal{N}(u(\xi,\eta)) + \Re(u(\xi,\eta))]\right], \tag{15}$$

where $\Theta(\xi,\eta)$ represents the contribution from the source term and I.C. To effectively address the problem, the HPSuTS introduces a novel technique for decomposing the solution $u(\xi,\eta)$ into an infinite series of components:

$$u(\xi,\eta) = \sum_{m=0}^{\infty} \mathrm{p}^m u_m(\xi,\eta). \tag{16}$$

Similarly, the non-linear term $\mathcal{N}(u(\xi,\eta))$ is decomposed as:

$$\mathcal{N}(u(\xi,\eta)) = \sum_{m=0}^{\infty} \mathrm{p}^m \mathcal{H}_m(u). \tag{17}$$

Here, the He's polynomials [18], denoted by $\mathcal{H}_m(u_0, u_1, \ldots, u_n)$, are defined as follows:

$$\mathcal{H}_m(u_0, u_1,, \ldots, u_n) = \frac{1}{m!} \frac{\partial^m}{\partial p^m} \left[ \mathcal{N}(\sum_{i=0}^{\infty} p^i u_i) \right]_{p=0}, \quad m = 0, 1, 2, 3, \ldots \tag{18}$$

With the utilization of the previously presented equations, the HPSuTS transforms the problem into a series of iterative equations for $u_m(\xi, \eta)$. Utilizing Equations (16) and (17), we elegantly transform Equation (15) to yield the following insightful expressions:

$$\sum_{m=0}^{\infty} p^m u_m(\xi, \eta) = \Theta(\xi, \eta) - p \left( \mathscr{S}^{-1} \left[ v^\alpha \mathscr{S} \left[ \Re \left( \sum_{m=0}^{\infty} p^m u_m(\xi, \eta) \right) + \mathcal{N} \left( \sum_{m=0}^{\infty} p^m \mathcal{H}_m(v) \right) \right] \right] \right). \tag{19}$$

Through a series of consecutive iterations and evaluation of well-established components of $p$, we uncover the following pattern:

$$p^0 : u_0(\xi, \eta) = \Theta(\xi, \eta)$$

$$p^1 : u_1(\xi, \eta) = -\mathscr{S}^{-1} \left[ v^\alpha \mathscr{S} \left[ \Re \left( u_0(\xi, \eta) \right) + \mathcal{N} \left( u_0(\xi, \eta) \right) \right] \right]$$

$$p^2 : u_2(\xi, \eta) = -\mathscr{S}^{-1} \left[ v^\alpha \mathscr{S} \left[ \Re \left( u_1(\xi, \eta) \right) + \mathcal{N} \left( u_1(\xi, \eta) \right) \right] \right] \tag{20}$$

$$p^3 : u_3(\xi, \eta) = -\mathscr{S}^{-1} \left[ v^\alpha \mathscr{S} \left[ \Re \left( u_2(\xi, \eta) \right) + \mathcal{N} \left( u_2(\xi, \eta) \right) \right] \right]$$

$$\vdots$$

Consequently, the solution to the intricate FDEs emerges as the sum of these terms:

$$u(\xi, \eta) = \sum_{m=0}^{\infty} u_n(\xi, \eta) = u_0(\xi, \eta) + u_1(\xi, \eta) + u_2(\xi, \eta) + \ldots \tag{21}$$

This method adeptly harnesses the concept of successive iterations and mathematical scrutiny to unveil a holistic solution for the complex FDEs, effectively showcasing the skills of mathematical techniques in tackling intricate equations.

In the forthcoming sections, we will explore the practical implementation of the HPSuTS and assess its efficacy in resolving NS-TFEs.

## 4. Implementation of the Proposed Technique

Let us consider the NS-TFEs in general form as [29,35]:

$$^C \mathcal{D}^\alpha u(r, \eta) = P + m \left( \mathcal{D}_{rr} u + \frac{1}{r} \mathcal{D}_r u \right), \quad 0 < \alpha \leq 1. \tag{22}$$

where $P = -\frac{\partial p}{\rho \partial z}$, $p$ denotes the pressure, m the kinematics viscosity, and $\rho$ is the density of a viscous fluid of unsteady flow. When $\alpha = 1$, the Equation (22) becomes the classical NSEs.

### 4.1. Case: 1

Let us first consider the NS-TFEs (22) [35] as:

$$^C \mathcal{D}^\alpha u(r, \eta) = P + \mathcal{D}_{rr} u + \frac{1}{r} \mathcal{D}_r u, \quad 0 < \alpha \leq 1. \tag{23}$$

with I.C:

$$u(r, 0) = 1 - r^2$$

By applying SuT on both sides of (23), the equation can be rewritten as:

$$\mathscr{S}[^{\mathcal{C}}\mathcal{D}^{\alpha}u(\mathrm{r},\eta)](\nu) = \mathscr{S}[P + \mathcal{D}_{\mathrm{rr}}u + \frac{1}{\mathrm{r}}\mathcal{D}_{\mathrm{r}}u](\nu). \tag{24}$$

By applying definition Section 2.6.5, the following expression is derived:

$$\mathscr{S}[u(\mathrm{r},\eta)](\nu) = u(\mathrm{r},0) + \nu^{\alpha}\mathscr{S}[P] + \nu^{\alpha}\mathscr{S}[\mathcal{D}_{\mathrm{rr}}u + \frac{1}{\mathrm{r}}\mathcal{D}_{\mathrm{r}}u]. \tag{25}$$

Incorporating the I.C, the above equation becomes:

$$\mathscr{S}[u(\mathrm{r},\eta)](\nu) = 1 - \mathrm{r}^2 + \nu^{\alpha}\mathscr{S}[P] + \nu^{\alpha}\mathscr{S}[\mathcal{D}_{\mathrm{rr}}u + \frac{1}{\mathrm{r}}\mathcal{D}_{\mathrm{r}}u]. \tag{26}$$

Finally, by taking the inverse SuT, we obtain:

$$u(\mathrm{r},\eta) = 1 - \mathrm{r}^2 + P\frac{\eta^{\alpha}}{\Gamma(\alpha+1)} + \mathscr{S}^{-1}\left[\nu^{\alpha}\mathscr{S}\left[\mathcal{D}_{\mathrm{rr}}u + \frac{1}{\mathrm{r}}\mathcal{D}_{\mathrm{r}}u\right]\right], \tag{27}$$

By implementing the HPS to the above equation we get:

$$\sum_{m=0}^{\infty}\mathrm{p}^m u_m(\mathrm{r},\eta) = 1 - \mathrm{r}^2 + P\frac{\eta^{\alpha}}{\Gamma(\alpha+1)} + \mathrm{p}\left(\mathscr{S}^{-1}\left[\nu^{\alpha}\mathscr{S}\left[\sum_{m=0}^{\infty}\mathrm{p}^m\mathcal{D}_{\mathrm{rr}}u_m + \frac{1}{\mathrm{r}}\sum_{m=0}^{\infty}\mathrm{p}^m\mathcal{D}_{\mathrm{r}}u_m\right]\right]\right) \tag{28}$$

The coefficients of the powers p are equated as follows:

$$\mathrm{p}^0 : u_0(\mathrm{r},\eta) = 1 - \mathrm{r}^2 + P\frac{\eta^{\alpha}}{\Gamma(\alpha+1)},$$

$$\mathrm{p}^1 : u_1(\mathrm{r},\eta) = \mathscr{S}^{-1}\left[\nu^{\alpha}\mathscr{S}\left[\mathcal{D}_{\mathrm{rr}}u_0 + \frac{1}{\mathrm{r}}\mathcal{D}_{\mathrm{r}}u_0\right]\right] = \frac{-4\eta^{\alpha}}{\Gamma(\alpha+1)},$$

$$\mathrm{p}^2 : u_2(\mathrm{r},\eta) = \mathscr{S}^{-1}\left[\nu^{\alpha}\mathscr{S}\left[\mathcal{D}_{\mathrm{rr}}u_1 + \frac{1}{\mathrm{r}}\mathcal{D}_{\mathrm{r}}u_1\right]\right] = 0,$$

$$\mathrm{p}^3 : u_3(\mathrm{r},\eta) = \mathscr{S}^{-1}\left[\nu^{\alpha}\mathscr{S}\left[\mathcal{D}_{\mathrm{rr}}u_2 + \frac{1}{\mathrm{r}}\mathcal{D}_{\mathrm{r}}u_2\right]\right] = 0,$$

and so on. By substituting the above values into the equation, we have:

$$u(\mathrm{r},\eta) = u_0(\mathrm{r},\eta) + u_1(\mathrm{r},\eta) + u_2(\mathrm{r},\eta) + u_3(\mathrm{r},\eta) + u_4(\mathrm{r},\eta) + \dots$$

$$= 1 - \mathrm{r}^2 + (P-4)\frac{\eta^{\alpha}}{\Gamma(\alpha+1)} + 0 + 0 + \dots \tag{29}$$

When $\alpha = 1$ the solution becomes,

$$u(\mathrm{r},\eta) = 1 - \mathrm{r}^2 + (P-4)\eta.$$

which is the required solution as in [29,35]. The approximate solutions for different values of $\alpha$ under two distinct scenarios i.e., for $P = 1$ and $P = 5$ are shown in the Figure 1a and 1b respectively. Moreover, the 3D representation of the solution is provided in Figure 2. Also, the tabular representation of the solution for various fractional order is provided in Tables 1 and 2 for the parameter $P = 1$ and $P = 5$ respectively.

**Table 1.** Approximate solution of (23) at $P = 1$.

| $\eta$ | $r$ | $\alpha = 0.7$ | $\alpha = 0.8$ | $\alpha = 0.9$ | $\alpha = 1$ |
|---|---|---|---|---|---|
| | 0.0 | 0.341235 | 0.489503 | 0.607308 | 0.700000 |
| | 0.2 | 0.301235 | 0.449503 | 0.567308 | 0.660000 |
| 0.1 | 0.4 | 0.181235 | 0.329503 | 0.447308 | 0.540000 |
| | 0.6 | −0.018764 | 0.129503 | 0.247308 | 0.340000 |
| | 0.8 | −0.298764 | −0.150496 | −0.032691 | 0.060000 |
| | 1.0 | −0.658764 | −0.510496 | −0.392691 | −0.300000 |

**Table 2.** Approximate solution of (23) at $P = 5$.

| $\eta$ | $r$ | $\alpha = 0.7$ | $\alpha = 0.8$ | $\alpha = 0.9$ | $\alpha = 1$ |
|---|---|---|---|---|---|
| | 0.0 | 1.219588 | 1.170165 | 1.130897 | 1.100000 |
| | 0.2 | 1.179588 | 1.130165 | 1.090897 | 1.060000 |
| 0.1 | 0.4 | 1.059588 | 1.010165 | 0.970897 | 0.940000 |
| | 0.6 | 0.859588 | 0.810165 | 0.770897 | 0.740000 |
| | 0.8 | 0.579588 | 0.530165 | 0.490897 | 0.460000 |
| | 1.0 | 0.219588 | 0.170165 | 0.130897 | 0.100000 |

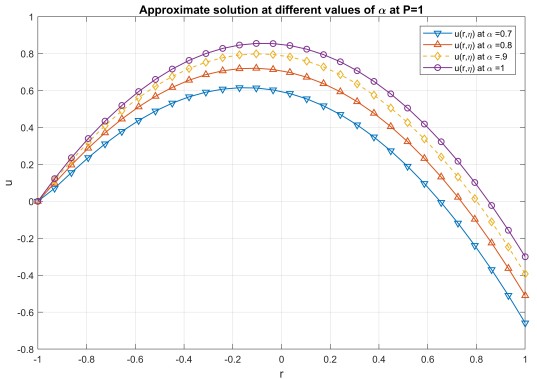 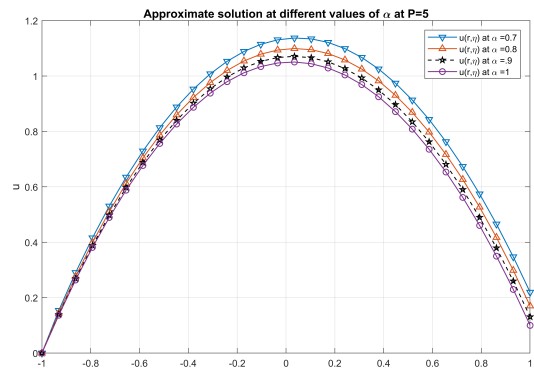

(**a**) Approximate Solution when P = 1      (**b**) Approximate Solution when P = 5

**Figure 1.** Numerical Comparison of solution.

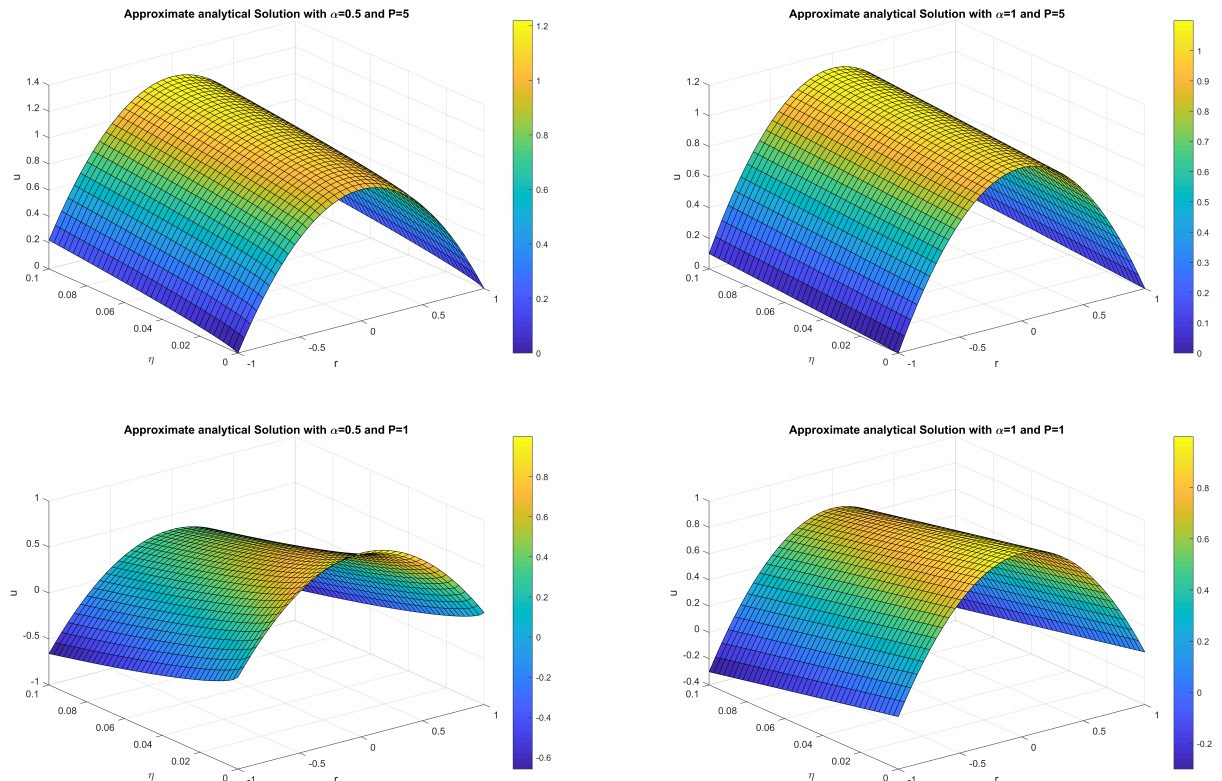

**Figure 2.** Approximate analytical solution.

*4.2. Case: 2*

Let us consider the NS-TFEs (22) [35] as:

$$^{\mathcal{C}}\mathcal{D}^{\alpha}u(\mathrm{r},\eta) = \mathcal{D}_{\mathrm{rr}}u + \frac{1}{\mathrm{r}}\mathcal{D}_{\mathrm{r}}u, \quad 0 < \alpha \leq 1. \tag{30}$$

with I.C:

$$u(\mathrm{r},0) = \mathrm{r}$$

By taking SuT on both sides of the above equations, the equations becomes:

$$\mathscr{S}[^{\mathcal{C}}\mathcal{D}^{\alpha}u(\mathrm{r},\eta)] = \mathscr{S}[\mathcal{D}_{\mathrm{rr}}u + \frac{1}{\mathrm{r}}\mathcal{D}_{\mathrm{r}}u]. \tag{31}$$

By applying definition Section 2.6.5, the following expression is derived:

$$\mathscr{S}[u(\mathrm{r},\eta)](v) = u(\mathrm{r},0) + v^{\alpha}\mathscr{S}[\mathcal{D}_{\mathrm{rr}}u + \frac{1}{\mathrm{r}}\mathcal{D}_{\mathrm{r}}u]. \tag{32}$$

Incorporating the I.C, the equation becomes:

$$\mathscr{S}[u(\mathrm{r},\eta)](v) = \mathrm{r} + v^{\alpha}\mathscr{S}[\mathcal{D}_{\mathrm{rr}}u + \frac{1}{\mathrm{r}}\mathcal{D}_{\mathrm{r}}u]. \tag{33}$$

Finally, by using the inverse SuT, we obtain:

$$u(\mathrm{r},\eta) = \mathrm{r} + \mathscr{S}^{-1}\left[v^{\alpha}\mathscr{S}\left[\mathcal{D}_{\mathrm{rr}}u + \frac{1}{\mathrm{r}}\mathcal{D}_{\mathrm{r}}u\right]\right], \tag{34}$$

By implementing the HPS to the above equation we get:

$$\sum_{m=0}^{\infty} \mathrm{p}^m u_m(\mathrm{r}, \eta) = \mathrm{r} + \mathrm{p}\left(\mathscr{S}^{-1}\left[\nu^\alpha \mathscr{S}\left[\sum_{m=0}^{\infty} \mathrm{p}^m \mathcal{D}_{\mathrm{rr}} u_m + \frac{1}{\mathrm{r}} \sum_{m=0}^{\infty} \mathrm{p}^m \mathcal{D}_{\mathrm{r}} u_m\right]\right]\right) \tag{35}$$

The coefficients of the powers p are equated as follows:

$$\mathrm{p}^0 : u_0(\mathrm{r}, \eta) = \mathrm{r},$$

$$\mathrm{p}^1 : u_1(\mathrm{r}, \eta) = \mathscr{S}^{-1}\left[\nu^\alpha \mathscr{S}\left[\mathcal{D}_{\mathrm{rr}} u_0 + \frac{1}{\mathrm{r}} \mathcal{D}_{\mathrm{r}} u_0\right]\right] = \frac{1}{\mathrm{r}} \frac{\eta^\alpha}{\Gamma(\alpha + 1)},$$

$$\mathrm{p}^2 : u_2(\mathrm{r}, \eta) = \mathscr{S}^{-1}\left[\nu^\alpha \mathscr{S}\left[\mathcal{D}_{\mathrm{rr}} u_1 + \frac{1}{\mathrm{r}} \mathcal{D}_{\mathrm{r}} u_1\right]\right] = \frac{1}{\mathrm{r}^3} \frac{\eta^{2\alpha}}{\Gamma(2\alpha + 1)},$$

$$\mathrm{p}^3 : u_3(\mathrm{r}, \eta) = \mathscr{S}^{-1}\left[\nu^\alpha \mathscr{S}\left[\mathcal{D}_{\mathrm{rr}} u_2 + \frac{1}{\mathrm{r}} \mathcal{D}_{\mathrm{r}} u_2\right]\right] = \frac{3^2}{\mathrm{r}^5} \frac{\eta^{3\alpha}}{\Gamma(3\alpha + 1)},$$

and so on. By substituting the above values into the equation, we have:

$$u(\mathrm{r}, \eta) = u_0(\mathrm{r}, \eta) + u_1(\mathrm{r}, \eta) + u_2(\mathrm{r}, \eta) + u_3(\mathrm{r}, \eta) + \ldots$$

$$= \mathrm{r} + \frac{1}{\mathrm{r}} \frac{\eta^\alpha}{\Gamma(\alpha + 1)} + \frac{1}{\mathrm{r}^3} \frac{\eta^{2\alpha}}{\Gamma(2\alpha + 1)} + \frac{3^2}{\mathrm{r}^5} \frac{\eta^{3\alpha}}{\Gamma(3\alpha + 1)} + \ldots \tag{36}$$

$$= \mathrm{r} + \sum_{m=1}^{\infty} \frac{1^2 \times 3^2 \times 5^2 \times \cdots \times (2m - 3)^2}{\mathrm{r}^{2m-1}} \frac{\eta^{m\alpha}}{\Gamma(m\alpha + 1)}.$$

When $\alpha = 1$ the solution becomes,

$$u(\mathrm{r}, \eta) = \mathrm{r} + \sum_{m=1}^{\infty} \frac{1^2 \times 3^2 \times 5^2 \times \cdots \times (2m - 3)^2}{\mathrm{r}^{2m-1}} \frac{\eta^m}{\Gamma(m + 1)}.$$

which is the required solution as in [29,35]. The approximate solutions for different values of $\alpha$ are shown in the Figures 3 and 4. Moreover, tabular representation of solution is provided in Table 3.

**Table 3.** Approximate solution of (30).

| $\eta$ | $r$ | $\alpha = 0.7$ | $\alpha = 0.8$ | $\alpha = 0.9$ | $\alpha = 1$ |
|---|---|---|---|---|---|
| | 0.1 | $3.3847e^4$ | $2.3455e^4$ | $1.5334e^4$ | $9.5311e^3$ |
| | 0.2 | 273.0877 | 190.5927 | 125.8219 | 79.2547 |
| 0.1 | 0.4 | 3.3648 | 2.5651 | 1.9244 | 1.4468 |
| | 0.6 | 1.1488 | 1.0235 | 0.9221 | 0.8426 |
| | 0.8 | 1.1102 | 1.0421 | 0.9872 | 0.9438 |
| | 1.0 | 1.2314 | 1.1805 | 1.1397 | 1.1074 |

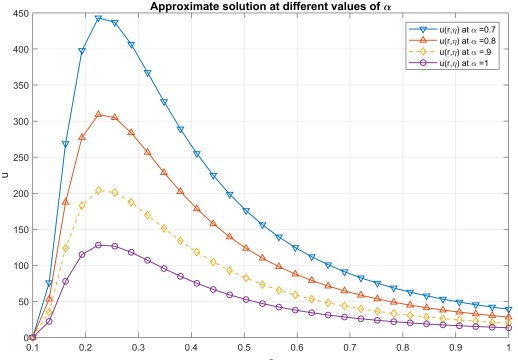

**Figure 3.** Numerical Comparison of Solution.

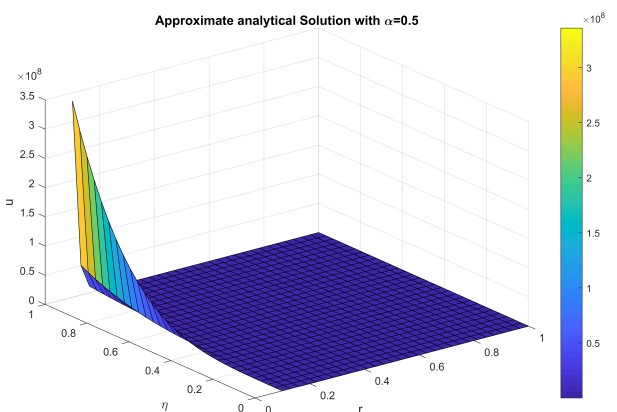 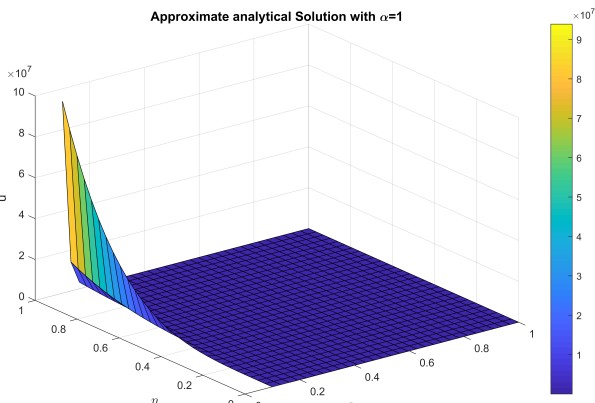

**Figure 4.** Approximate analytical solution.

## 5. Conclusions

This study introduces the HPSuT method, which is a semi-analytical technique used to solve NS-TFEs. By combining the SuT and HPS techniques, we were able to successfully solved the NS-TFEs and obtained an approximate solution. The proposed solution is supported by graphical results in both 2D and 3D, which depict the solution for different values of $\alpha$. Furthermore, numerical results obtained using the suggested technique for various values of $\alpha$ are compared, demonstrating that as the value transitions from a fractional order to an integer order, the solution becomes increasingly similar to the exact solution. The results presented in Tables 1–3 also indicate that the series solution rapidly converges to the exact solution as the non-integer order approaches an integer order.

The HPSuT method offers several advantages that contribute to its effectiveness and interactivity. Firstly, it provides series solutions that converge quickly and exhibit a high degree of accuracy, enabling reliable numerical results to be efficiently obtained. Additionally, this method significantly reduces the computational workload compared to other traditional methods, which is particularly beneficial for complex problems and large datasets. Moreover, the HPSuT method is not limited to specific types of FDEs, as it can be applied to both linear and non-linear FDEs. This versatility makes it a valuable tool applicable in various scientific and engineering fields.

To summarize, the HPSuT method has been demonstrated to be an interactive and efficient approach for solving NS-TFEs. Its accuracy, computational efficiency, and applicability to different types of FDEs make it a valuable tool for researchers and practitioners in various domains.

**Author Contributions:** Conceptualization, S.I. and F.M.; methodology, S.I.; software, S.I.; validation, S.I. and F.M.; formal analysis, S.I. and F.M.; investigation, S.I. and F.M.; writing—original draft preparation, S.I.; writing—review and editing, S.I. and F.M.; visualization, S.I and F.M.; supervision, S.I. and F.M. All authors have read and agreed to the published version of the manuscript.

**Funding:** This research received no external funding.

**Institutional Review Board Statement:** Not applicable.

**Informed Consent Statement:** Not applicable.

**Conflicts of Interest:** The authors declare no conflict of interest.

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
