# Peer review of "An Approach for Approximating Analytical Solutions of the Navier-Stokes Time-Fractional Equation Using the Homotopy Perturbation Sumudu Transform’s Strategy"

_axioms, doi:10.3390/axioms12111025_

Round 1

Reviewer 1 Report

Comments and Suggestions for Authors

The paper deals with the Sumudu transform’s properties to tackle the time fractional differential equation. Authors present a comprehensive formulation of a novel approach termed ‘Homotopy Perturbation Sumudu Transform Strategy’ (HPSuTS).  This approach is applied to address complex FDE, которое авторы называют time fractional Navier Stokes equation. The article contributes new scientific results that, apparently, may be of interest to specialists, but we have some comments about their presentation.

1. The reference list is too brief (only 17 works) and doesn’t reflect the state-of-art in the field of approximate methods for solving FDEs. Such a large number of works on this topic is known. Therefore, choosing the most relevant publications seems to be a difficult task. Let us entrust its solution to the authors of the paper.

2. Bibliographic references are made carelessly and with misprints (see [3, 11]).

3. The 'Navier-Stokes system' is well known in the scientific literature, but the term 'fractional Navier Stokes equation' is not common. It would be helpful for readers to get detailed explanations regarding the choice of the term, its correctness and the relationship between the 'Navier-Stokes System' and the 'fractional Navier Stokes equation'.

4. It is very good that the description of the approximate method is accompanied by numerical calculations, but they are uninformative. There is no error analysis or description of the results. There are very few examples, although the format of the journal allows them to be given almost without restrictions.

5. The conclusion is unsatisfactory. It consists of five lines and doesn't draw conclusions, discusses of the results, or points out possible directions for future research.

Comments on the Quality of English Language

In my opinion, the quality of English language is acceptable.

Author Response

Authors' response to Reviewer 1

Dear Reviewer,

The authors express profound appreciation for the reviewer´s comments and suggestions. They were helpful in making the revised manuscript much stronger and much focused. (Please note: {All the track changers below are highlighted in red in the revised manuscript.})

Below, we respond specifically to your comments and suggestions.

(i) The reference list is too brief (only 17 works) and doesn’t reflect the state-of-art in the field of approximate methods for solving FDEs. Such a large number of works on this topic is known. Therefore, choosing the most relevant publications seems to be a difficult task. Let us entrust its solution to the authors of the paper.

Answer: Your suggestion is very appropriate. The number of references has increased significantly, so that the introduction can better reflect the state of the art of the field in which our research is located. (See revised version).

(ii) Bibliographic references are made carelessly and with misprints (see [3, 11]).

Answer: Thanks for your suggestion. We have carefully reviewed the references and corrected the indicated typos. (See revised version).

(iii) The 'Navier-Stokes system' is well known in the scientific literature, but the term 'fractional Navier Stokes equation' is not common. It would be helpful for readers to get detailed explanations regarding the choice of the term, its correctness and the relationship between the 'Navier-Stokes System' and the 'fractional Navier Stokes equation'.

Answer: Your suggestion is very appropriate. In the introduction we have included a new paragraph that considers your suggestion. (See revised version).

(iv) It is very good that the description of the approximate method is accompanied by numerical calculations, but they are uninformative. There is no error analysis or description of the results. There are very few examples, although the format of the journal allows them to be given almost without restrictions.

Answer: Your suggestion is very appropriate. On the one hand, we believe that the two examples included in the work sufficiently illustrate the proposed method, since we have tried to present the developments carried out in detail. However, we have considered your suggestion by including in the revised version of the work some tables that illustrate the approximate solution of the NS-TFE studied, which complete the information provided by the figures. (See revised version).

(v) The conclusion is unsatisfactory. It consists of five lines and doesn't draw conclusions, discusses of the results, or points out possible directions for future research.

Answer: Your suggestion is very appropriate. We have tried to improve the Conclusions in the direction you indicate. (See revised version).

(vi) Finally, we would like to indicate that attempts have been made to improve English when needed.

Reviewer 2 Report

Comments and Suggestions for Authors

Comments to the authors: Review of “An Approach for Approximating Analytical Solutions of the Navier-Stokes Time-Fractional Equation using the Homotopy Perturbation Sumudu Transform’s Strategy”.

I have read the manuscript in a detailed fashion. This paper is interesting and might be a significant addition to the existing literature. However, the paper can be considered after the authors make the following modifications:

  1. The originality of the paper needs to be stated clearly. It is important to have sufficient results to justify the novelty of a high-quality journal paper. The introduction should make a compelling case for why the study is useful, along with a clear statement of its novelty or originality, by providing relevant information and providing answers to basic questions such as: What is already known in the open literature? What is missing (i.e., research gaps)? What needs to be done, why, and how? Clear statements of the novelty of the work should also appear briefly in the Abstract and Conclusions sections.
  2. What is your contribution? It should be marked clearly in the abstract and conclusion in detail.
  3. What is the purpose of Sumudu Transform in this study? Does Sumudu Transform have more advantage over the well-known Laplace Transform?
  4. What is the purpose of using fractional derivatives in this study?
  5. In page 5, equation in subsection 2.6.5 should have f(0^+) instead of f(0+).
  6. The whole manuscript should be checked for typos and grammatical errors. An overall review is needed to fix the grammatical and typographical errors in the manuscript.
  7. For a better presentation of the paper, the authors should discuss and cite recent works given as:

a)     Homotopy perturbation Shehu transform method for solving fractional models arising in applied sciences. Journal of Applied Mathematics and Computational Mechanics, 20(1), 71-82 (2021).

b)     Modified homotopy methods for generalized fractional perturbed Zakharov–Kuznetsov equation in dusty plasma. Advances in Difference Equations, 2021(1), 1-27 (2021).

88.     All references that have been cited throughout the text should be checked very carefully.

Comments on the Quality of English Language

Moderate editing of English language required.

Author Response

Authors' response to Reviewer 2

Dear Reviewer,

The authors express profound appreciation for the reviewer´s comments and suggestions. They were helpful in making the revised manuscript much stronger and much focused. (Please note: {All the track changers below are highlighted in red in the revised manuscript.})

Below, we respond specifically to your comments and suggestions.

(i) The originality of the paper needs to be stated clearly. It is important to have sufficient results to justify the novelty of a high-quality journal paper. The introduction should make a compelling case for why the study is useful, along with a clear statement of its novelty or originality, by providing relevant information and providing answers to basic questions such as: What is already known in the open literature? What is missing (i.e., research gaps)? What needs to be done, why, and how? Clear statements of the novelty of the work should also appear briefly in the Abstract and Conclusions sections.

Answer: Your suggestion is very appropriate. In the Abstract, Introduction, and Conclusion sections we have included new paragraphs or sentences that consider your suggestion. (See revised version).

(ii) What is your contribution? It should be marked clearly in the abstract and conclusion in detail.

Answer: Your suggestion is very appropriate. In the Abstract, Introduction and Conclusion sections we have included new paragraphs or sentences that consider your suggestion. (See revised version).

(iii) What is the purpose of Sumudu Transform in this study? Does Sumudu Transform have more advantage over the well-known Laplace Transform?

Answer: Your suggestion is very appropriate, especially if we think about the potential readership of the paper. In this sense, we have included in the introduction some sentences that answer these questions. (See revised version).

(iv) What is the purpose of using fractional derivatives in this study?

Answer: Your suggestion is very appropriate, especially if we think about the potential readership of the paper. In this sense, we have included in the introduction some sentences that answer these questions. (See revised version).

(v) In page 5, equation in subsection 2.6.5 should have f(0^+) instead of f(0+).

Answer: Thanks for your suggestion. We have corrected the indicated typo. (See revised version).

(vi) The whole manuscript should be checked for typos and grammatical errors. An overall review is needed to fix the grammatical and typographical errors in the manuscript.

Answer: Thanks for your suggestion. A general review of the manuscript has been carried out and all grammatical and typographical errors detected have been corrected. Likewise, we have tried to improve English when needed.

(vii) For a better presentation of the paper, the authors should discuss and cite recent works given as:
a) Homotopy perturbation Shehu transform method for solving fractional models arising in applied sciences. Journal of Applied Mathematics and Computational Mechanics, 20(1), 71-82 (2021).
b) Modified homotopy methods for generalized fractional perturbed Zakharov–Kuznetsov equation in dusty plasma. Advances in Difference Equations, 2021(1), 1-27 (2021).

Answer: Your suggestion is very appropriate. These interesting investigations have been included, both in Section 1. Introduction and in the references. (See in the revised version).

(viii) All references that have been cited throughout the text should be checked very carefully.

Answer: Thanks for your suggestion. All references cited throughout the paper have been carefully reviewed.

Reviewer 3 Report

Comments and Suggestions for Authors

the paper discusses on time fractional Navier Stokes equation, in the menanig of approximation solutions.

The abstract is clear and well written.

The introduction gives a survey on fractional calculus, but it could be useful to draw some different approaches that are in literature as for example contained in the following paper:

Wael W. Mohammed, Farah M. Al-Askar, Clemente Cesarano, Mahmoud El-Morshedy (2023). Solitary Wave Solution of a Generalized Fractional–Stochastic Nonlinear Wave Equation for a Liquid with Gas Bubbles. MATHEMATICS, vol. 11, p. 1-14

Sections 2, 3 and 4 are clear and well written.

Section 5, conclusions, must be enlarged: more comments on the results presented and some remarks on future researches must be added.

Author Response

Authors' response to Reviewer 3

Dear Reviewer,

The authors express profound appreciation for the reviewer´s comments and suggestions. They were helpful in making the revised manuscript much stronger and much focused. (Please note: {All the track changers below are highlighted in red in the revised manuscript.})

Below, we respond specifically to your comments and suggestions.

(i) The introduction gives a survey on fractional calculus, but it could be useful to draw some different approaches that are in literature as for example contained in the following paper:
Wael W. Mohammed, Farah M. Al-Askar, Clemente Cesarano, Mahmoud El-Morshedy (2023). Solitary Wave Solution of a Generalized Fractional–Stochastic Nonlinear Wave Equation for a Liquid with Gas Bubbles. MATHEMATICS, vol. 11, p. 1-14

Answer: Your suggestion is very appropriate. This interesting research has been included, both in Section 1. Introduction and in the references. (See in the revised version).

(ii) Section 5, conclusions, must be enlarged: more comments on the results presented and some remarks on future researcher must be added.

Answer: Your suggestion is very appropriate. We have tried to improve the Conclusions in the direction you indicate. (See revised version).

(iii) Finally, we would like to indicate that attempts have been made to improve English when needed.

Round 2

Reviewer 1 Report

Comments and Suggestions for Authors

The authors did a good job, and the paper became much better. I suppose it can be accepted.

Comments on the Quality of English Language

In my opinion, the quality of English language is acceptable.

Reviewer 2 Report

Comments and Suggestions for Authors

In this new revision of the article, the authors have presented a detailed answer to my previous comments and in my opinion, the quality of the present work has been improved considerably. Therefore, I would like to recommend this article for its publication in "Axioms" because the authors have answered all my previous questions.

Comments on the Quality of English Language

Minor editing of English language required.